# Biomaterials as Implants in the Orthopedic Field for Regenerative Medicine: Metal versus Synthetic Polymers

**DOI:** 10.3390/polym15122601

**Published:** 2023-06-07

**Authors:** Faisal Dakhelallah Al-Shalawi, Azmah Hanim Mohamed Ariff, Dong-Won Jung, Mohd Khairol Anuar Mohd Ariffin, Collin Looi Seng Kim, Dermot Brabazon, Maha Obaid Al-Osaimi

**Affiliations:** 1Department of Mechanical and Manufacturing Engineering, Faculty of Engineering, Universiti Putra Malaysia, Serdang 43400, Selangor, Malaysia; faisalalshe22@gmail.com (F.D.A.-S.); khairol@upm.edu.my (M.K.A.M.A.); 2Research Center Advanced Engineering Materials and Composites (AEMC), Faculty of Engineering, Universiti Putra Malaysia, Serdang 43400, Selangor, Malaysia; 3Faculty of Applied Energy System, Major of Mechanical Engineering, Jeju National University, 102 Jejudaehak-ro, Jeju-si 63243, Republic of Korea; 4Department of Orthopaedic, Faculty of Medicine and Health Sciences, Universiti Putra Malaysia, Serdang 43400, Selangor, Malaysia; collinlooi@upm.edu.my; 5Advanced Manufacturing Research Centre, and Advanced Processing Technology Research Centre, School of Mechanical and Manufacturing Engineering, Dublin City University, Glasnevin, D09 V209 Dublin 9, Ireland; dermot.brabazon@dcu.ie; 6Department of Microbiology, Faculty of Agriculture, Universiti Putra Malaysia, Serdang 43400, Selangor, Malaysia; moas22@windowslive.com

**Keywords:** orthopedic, bone, biodegradable, corrosion resistance, biocompatibility

## Abstract

Patients suffering bone fractures in different parts of the body require implants that will enable similar function to that of the natural bone that they are replacing. Joint diseases (rheumatoid arthritis and osteoarthritis) also require surgical intervention with implants such as hip and knee joint replacement. Biomaterial implants are utilized to fix fractures or replace parts of the body. For the majority of these implant cases, either metal or polymer biomaterials are chosen in order to have a similar functional capacity to the original bone material. The biomaterials that are employed most often for implants of bone fracture are metals such as stainless steel and titanium, and polymers such as polyethene and polyetheretherketone (PEEK). This review compared metallic and synthetic polymer implant biomaterials that can be employed to secure load-bearing bone fractures due to their ability to withstand the mechanical stresses and strains of the body, with a focus on their classification, properties, and application.

## 1. Introduction

Biomaterials have been utilized for the treatment of human diseases since ancient times. For example, as early as 2000 BC, the Egyptians used ivory to replace lost teeth [1], [2], and employed wood to replace missing bones such as legs and toes [3]. They also used braces and splints to support and protect fractured bones after surgical procedures [4]. During the same period, copper was used to replace missing bony parts of the human body, but these implants failed due to the toxic effects of copper ions. The ancient Indian text from the Vedic period (1800–1500 BC) mentioned the use of teeth, eyes, and artificial legs. Autogenous tissues, or tissues from the patient’s own body, were also employed during this period to replace missing body parts [3].

Biomaterials have become increasingly crucial in modern times, serving various applications as a result of advances in medicine and material processing in recent decades. They are extensively utilized in various fields, including orthopedics, dentistry, cardiovascular devices, drug delivery, and skin tissue engineering. These materials are specifically designed to interact with biological systems to repair, assess, or replace damaged or malfunctioning bodily tissues, organs, or systems [2,5,6,7], as shown in Figure 1. They are engineered for medical use either independently or as part of a biocompatible system with the body’s tissues and organs [8,9].

Biocompatibility is a term commonly used in biomaterials science to describe the interactions between foreign matter and the body [11]. Specifically, a substance’s biocompatibility is evaluated based on its ability to perform its intended medical function while eliciting an appropriate response from the host in a given application. It also encompasses the substance’s ability to interact with living systems without triggering any adverse reactions such as immune rejection, toxicity, or infection. Ultimately, the biomaterials must not produce any unwanted or unsuitable local or systemic effects [12,13]. Two major factors determine the biocompatibility of material: (i) the host’s reaction induced by the biomaterial and (ii) the degradation of the substance in the body’s environment. Often, both factors should be considered. One of the prime factors controlling biocompatibility is the material’s resistance to corrosion, which impacts the mechanical characteristics of biomaterials such as the specific weight and elastic modulus [5].

The majority of commercially utilized bio-implant materials are conventionally fabricated permanent metals and their alloys, such as stainless steel, titanium, and Co-Cr alloys [14]. They offer a stress-protective impact despite having excellent mechanical strength, biocompatibility, and acceptable wear resistance qualities [15,16] because the elasticity modulus of these bio-implants differs significantly from that of normal bone [17,18,19]. As a result, the bone can bear a much lower load, which progressively causes re-fracture and implant loosening as well as a considerable reduction in bio-efficacy. In addition, when the fracture is healed, a second surgery is necessary to remove the implant [14,20], causing colossal suffering to the patient. In addition, such an operation is quite expensive. According to studies, these follow-up surgeries, which are performed to remove permanent implants, account for approximately 30% of all orthopedic surgical procedures [14,21].

Corrosion plays a significant role in the design and selection of metals and alloys for use in vivo. During corrosion processes, allergenic, toxic/cytotoxic, or carcinogenic species such as Ni, Co, and Cr, may be released into the body. Moreover, different mechanisms of corrosion can contribute to implant loosening and failure [22,23,24]. Consequently, bio-implants often undergo corrosion and/or solubility testing before receiving approval from regulatory bodies [5].

Given the intricate nature of biomaterials and the significant responsibility involved in their use, a multidisciplinary approach is necessary for their design and development. Therefore, the involvement of professionals from diverse fields, such as chemists, biologists, engineers, histopathologists, and surgeons, is crucial to achieve the desired outcomes that would benefit patients with various pathologies [25].

Biomaterials are categorized into several types, including metals (e.g., stainless steel, titanium, gold, iron, magnesium), polymers (e.g., PLLA, PGA, PDS, nylon, silicone, polyester), ceramics (e.g., hydroxyapatite, alumina, zirconia), and composites, which combine materials from the above-mentioned categories [2,6]. Furthermore, they can be categorized into two main groups: (i) biodegradable materials that degrade and are absorbed into the surrounding tissue after implantation, and (ii) non-biodegradable materials that do not degrade or become absorbed. Inactive biomaterials exhibit limited or no interaction with tissues, while bioactive substances promote interactions with surrounding tissues. Degradable materials gradually release their mass into the surrounding tissues and may eventually disintegrate. Metals are generally inert; ceramics can be absorbable, inert, or bioactive; and polymers can be either inert or absorbable [26]. In tissue engineering applications, biomaterials can be further divided into two major classes based on their applications in hard tissues (such as bone, teeth, cartilage, and nails) and soft tissues (including skin, synovial membranes, ligaments, and fibrous tissues), with or without mineral constituents. Additionally, commonly used biomaterials can be classified as permanent or temporary implants depending on the required surgical fixation [2,27], as shown in Figure 2.

Metallic, polymeric, and ceramic materials contribute significantly to advancing orthopedic disease treatment. Since calcium phosphate ceramics have numerous disadvantages, such as mechanical weakness and poor resistance to crack growth, they can only be used in non-load-bearing applications [20,28]. For alumina- and zirconia-based bio-ceramics in high-load-bearing applications, static and cyclic fracture, the phenomena of slow crack growth, poor toughness, loss of toughness over time, stress corrosion, and susceptibility to tensile stresses are all major causes of worry; Therefore, designs have to be constrained concerning confined tensile loads or compressive loading [29,30]. Moreover, due to these issues, ceramics have not yet been utilized for fracture fixing [30]. Consequently, this review systemically compared the characteristics of biodegradable and non-biodegradable metals and synthetic polymers. Their potential for load-bearing bone fixations was evaluated in terms of biodegradability, mechanical properties, and biocompatibility.

## 2. Biomaterials in Orthopedics

Bone is a living tissue that serves as a natural complex, consisting of approximately 30% matrix, 60% minerals, and 10% water [2,31]. It serves multiple functions in the body, including providing mechanical support as the skeletal structure; serving as attachment sites for muscles, ligaments, and tendons; and protecting vital organs [32]. Bone also acts as a mineral store, contributing to blood production and overall bone health, which plays a crucial role in maintaining overall well-being [33,34]. 

Bone fractures are a prevalent type of traumatic injury globally, and their treatment often imposes a significant economic burden on society [35,36]. Despite the availability of advanced therapeutic strategies, complications such as delayed fracture healing or non-union occur in approximately 10% of cases, leading to prolonged recovery periods and increased hardship for patients. High-risk groups, such as those with osteoporosis, the elderly, malnourished individuals, post-menopausal women, or those with an impaired blood supply, are particularly susceptible to developing fracture healing disorders [35]. Additionally, individuals with joint diseases such as osteoarthritis and rheumatoid arthritis may require surgical interventions such as hip and knee joint replacements. Temporary fracture fixation appliances and components such as plates, screws, wires, and nails are among the orthopedic implants used in these cases [37]. Table 1 illustrates the locations of various human body fractures that demand some form of temporary fixture until they heal.

Continuous advancements are being made in the field of orthopedic implants to improve their interaction with the surrounding bone tissue, leading to positive and satisfactory outcomes for patients. The advantageous biological interaction between the implant and the surrounding bone is influenced by physical, mechanical, and topological characteristics [2,39]. Optimal fracture healing relies on achieving complete fixation of the fracture and promoting the preservation and growth of bony segments through local vascular remodeling [2,40]. To better understand the processes of adequate or impaired fracture healing, various in vivo, ex vivo, or in vitro models are available, providing opportunities for targeted and focused scientific research in both basic and translational settings [41].

Biomaterials are used in orthopedics to restore the structural integrity of damaged bone or to substitute it. Every biomaterial must fulfill multiple essential criteria, such as possessing suitable mechanical properties (e.g., precise weight and elastic modulus), demonstrating good biostability (resistance to hydrolysis, oxidation, and corrosion), ensuring biocompatibility, particularly in the case of bone prostheses (promoting osseointegration), exhibiting high bio-inertness (non-toxic and non-irritant characteristics), demonstrating high wear resistance, and enabling easy application in practical settings [30,33,42,43]; see Figure 3.

The mechanical demands of an implant are determined by its intended medical function, with factors such as strength for bearing loads and elasticity for withstanding shear stress playing a crucial role. In orthopedics, implant materials must withstand repetitive loading and unloading cycles under various forces such as bending, twisting, and shearing stress. Additionally, implant devices are exposed to corrosive environments over extended periods, potentially impacting their properties. Therefore, it is essential to accurately evaluate the mechanical properties of these materials to ensure fracture reduction and maintain optimal performance [44]. The assessment of mechanical properties involves examining the deformation (strain) generated by an applied force (stress). This evaluation provides valuable insights into the material’s ability to withstand and adapt to external forces, guiding the design and selection of implants that can meet the mechanical requirements of their intended applications [45].

In terms of biocompatibility, most metals employed as biomaterials have a relatively low intrinsic osteogenic and osteoimmune modulating potential [46], especially when compared to polymers such as polylactic acid (PLA). The presence of metallic materials as foreign bodies in the human body can often be a dangerous factor leading to chronic inflammation [47]. If these permanent implants are not promptly removed, they can cause serious allergic issues due to the accumulation of ions around the fracture site, resulting in osteolysis and impeding the formation of new bone [14]. In addition to metallic biomaterials, polymeric biomaterials are also employed for load-bearing applications. Several polymeric biomaterials have garnered significant attention in this field due to their ability to endure significant physiological stresses without fracturing or distorting [48]. In regenerative medicine applications, synthetic biopolymers offer several advantages over other non-biodegradable materials. They can be produced with consistently high quality and purity and can be shaped into various forms with desired bulk and surface properties [2,49]. Table 2 illustrates the mechanical characteristics, corrosion resistance, and biocompatibility of metallic biomaterials widely used in the medical field.

### 2.1. Metals and Alloys

Metallic biomaterials are essential for repairing or replacing damaged bone tissue due to their high mechanical strength and fracture toughness, making them better suited for load-bearing applications than ceramics or polymeric materials. In recent years, non-biodegradable metals such as titanium, titanium alloys, stainless steel, nitinol (nickel–titanium alloys), and cobalt-based alloys have been the most widely used biomaterials for medical implant devices [66].

#### 2.1.1. Non-Biodegradable Metals

##### Stainless Steel and Its Alloys

Since the 1930s, stainless steel (SS) has been a commonly used material for creating bone fixation plates. Stainless steel refers to a range of iron-based alloys that contain a significant amount of chromium (11–30wt %) and varying levels of nickel [67]. This versatile material is favored for its exceptional mechanical properties and cost-effectiveness when compared to other metals such as titanium alloys. Even today, stainless steel remains a crucial choice for temporary devices such as bone plates, fixating screws, and permanent orthopedic implants due to its widespread availability and desirable characteristics [68].

Compared to conventional steel, specifically 316 L austenitic stainless steel, stainless steel offers superior corrosion resistance. This property makes it a suitable choice for manufacturing prosthetic joints and bone plates [69]. Austenitic stainless steels, including 316 L, exhibit high stability and are less prone to hydrogen embrittlement compared to low-alloy steels, carbon steels, and less stable austenitic stainless steels. The presence of internal hydrogen enhances the yield and tensile strength while reducing ductility in the more stable 316 L austenitic stainless steels. The effect of hydrogen on fatigue properties depends on the internal hydrogen content [70].

##### Co–Cr Alloys

Cobalt–chromium-based alloys have significantly higher hardness and strength and better wear and corrosion resistance than Ti alloys. Co–Cr-based alloys for biomedical applications have previously been principally fabricated by molding and milling manners; However, portions fabricated by these traditional techniques required extra processing because of their poor dimensional and shaped precision [71].

Co–Cr–Mo alloy is one of the most vastly employed biomaterials for manufacturing orthopedic implants because of its superb compound of corrosion resistance, biocompatibility, and mechanical properties [72,73]. Nevertheless, Co–Cr–Mo alloy has lower automation, and it is challenging to manufacture some complicated orthopedic portions such as synthetic vertebrae utilizing the usual manufacturing manner [74].

##### Titanium (Ti) and Its Alloys

Titanium and its alloys have become the most widely used materials in the manufacture of biomaterial implants for fixing bone fractures due to their high biocompatibility and excellent mechanical properties [75,76]. However, the biological inertness of titanium alloys often results in poor and/or delayed osseointegration [77]. Two approaches are ordinarily commonly utilized to ameliorate the bioactivity of titanium-based alloys by targeting the substance’s compositional design or surface functionality [78].

The design of synthetic materials has great potential in fabricating bio-implants with incorporated structural steadiness and a desired biological function, which can help to avoid matrix coating interface problems or the mismatch of its properties due to surface operation [79]. In the evolution of biocompatible titanium alloys, biotoxicity and a low modulus of elasticity are important considerations. Some researchers have explored Ti–Al–V-based alloys with an extremely low modulus, but these alloys release toxic ions such as aluminum (Al) and vanadium (V), which can have long-term health effects [80]. The use of vanadium (V) compounds, for example, has been associated with DNA damage in blood cells, altered neurobehavioral functions, weight loss, and various toxicities [81,82]. Conversely, researchers have investigated a wide range of titanium alloys with different elemental compositions to achieve compatibility with various bone types. Parameters such as a low elastic modulus, non-toxicity, and biocompatibility have been extensively studied in the literature. Magnesium (Mg) is particularly renowned for its biodegradability.

Ti–Mg has emerged as a promising candidate for orthopedic implants due to its desirable characteristics, including a low elastic modulus, high strength-to-weight ratio, and good biocompatibility [83]. Currently, beta-type titanium alloys are being successfully used in orthopedic implants. These alloys are renowned for their very low Young’s modulus, high biocompatibility, excellent physical properties, and non-toxic nature, meeting the requirements for orthopedic applications. They also contain β-stabilizing elements such as Nb, Ta, Zr, Mo, and Hf [84]. Table 3 summarizes the advantages and disadvantages of various metallic materials used in orthopedic applications, along with their respective applications.

#### 2.1.2. Biodegradable Metals

Over the past two decades, there has been significant research and exploration of biodegradable metallic implants, specifically those composed of magnesium (Mg) and its alloys, iron (Fe) and its alloys, as well as zinc (Zn) and its alloys. These materials have garnered considerable interest due to their ability to degrade naturally, making them promising candidates for orthopedic implant applications [86,87,88,89,90,91,92,93,94,95,96,97].

##### Magnesium and Its Alloys

Magnesium and its alloys are considered promising biomaterials for orthopedic devices due to their excellent mechanical properties, biodegradability, and biocompatibility with human physiology [98]. However, one challenge associated with magnesium alloys is their susceptibility to corrosion in a biological environment. The significant corrosion rate and low bioactivity of magnesium implants pose a challenge that needs to be addressed before they can be used in clinical applications [99].

Magnesium alloys possess a similar elasticity to human bone, mitigating the negative effects of stress shielding in bone composition. Consequently, they are currently employed as temporary implants in the field of biomaterials. Within the biological environment, these alloys undergo complete degradation and are gradually substituted by newly regenerated bone, obviating the requirement for surgical intervention to extract the implant. This characteristic makes them highly desirable for metallic biomaterial implants used in bone regeneration applications that require temporary support [100].

However, there is a drawback: magnesium alloys degrade rapidly in the biological environment, necessitating effective control of the corrosion rate through bone tissue regeneration processes [101]. The rapid corrosion process can have adverse effects on the implant, including a decline in mechanical properties and the release of toxic by-products due to side reactions and corrosion accumulation.

As a result, there are significant implications in terms of cost and the overall health of the patient. Therefore, it is crucial to develop corrosion-resistant magnesium alloys in order to address these concerns in medical applications [100].

##### Fe and Its Alloys

Fe-based biodegradable materials as new-generation orthopedic implants draw increasing attention owing to their controllable internal pore structure, excellent mechanical characteristics, customizable complicated geometry, appropriate biocompatibility, and self-degradation feature [102,103]. In comparison to Mg-based alloys, pure Fe and its alloys possess high mechanical strength without causing hydrogen release during degradation. However, their degradation rates are considered too slow to align with bone growth, which is an issue requiring immediate attention [102,104,105,106,107,108,109].

Furthermore, Fe is a vital and scarce element in living organisms, playing a critical role in various physiological functions, including the formation, transportation, and storage of oxygen through hemoglobin, as well as the reduction in dinitrogen and ribonucleotides, and DNA installation [105,108,110,111,112,113]. Due to their combination of excellent strength and moderate corrosion rates, Fe and Fe-based bio-implants have been recognized as promising options for potential bone replacement or osteosynthesis materials [103,104,105,106,107]. Compared to Mg and its alloys, and Zn and its alloys, Fe combines superb compressive properties and tensile quality (compressive strength of 752 ± 13 MPa; tensile yield strength of 135 ± 15 MPa) [114,115]. These impressive mechanical characteristics make Fe well-suited for load-bearing devices, as highlighted in Table 4 [116].

However, the slow degradation rate of pure Fe in osteogenic environments (0.16 mm/year) [105] necessitates the incorporation of additional elements such as Mn, C, Si, Zn, and Pd. This strategic addition aims to improve the bio-absorption rate of iron-based materials and reduce their magnetic susceptibility, thereby enhancing their practicality for biodegradable applications [104,105,115,117,118,119]. Previous research has shown that the bone-forming capacity of Fe is similar to that of bio-inert stainless steel, indicating the need for improvement in the surface bioactivity of iron [115,120].

##### Zinc and Its Alloys

Over the past two decades, the development of biodegradable metal implants has predominantly centered around two types of metals: iron-based alloys and magnesium-based alloys. Iron-based alloys, despite their remarkable biocompatibility and high mechanical characteristics [117,118,121,122], are prone to significant corrosion, leading to the formation of a bulky iron oxide layer that can trigger inflammation [123]. On the other hand, magnesium-based alloys exhibit excellent biocompatibility [124,125], but the release of hydrogen gas during corrosive degradation can result in tissue separation and, in severe cases, gas embolism [126,127].

The limitations associated with magnesium and iron as biodegradable implant materials have prompted researchers to explore zinc and its alloys as alternatives. Zinc, being an essential trace element in the human body, offers acceptable corrosion rates and biocompatibility, making it suitable for orthopedic and other medical applications, including cardiovascular interventions; Zinc plays a critical role in various physiological processes such as nucleic acid metabolism, gene expression, and signal transduction [122,128,129,130,131]. However, it is important to note that excessive levels of zinc in the body can have detrimental effects, impairing normal growth and causing anemia by interfering with iron absorption [132,133].

Although laboratory studies have shown promising results for zinc-based materials as biodegradable implants [134,135,136], their performance in vivo remains significantly uncertain [137,138]. However, in vivo investigations have revealed that pure zinc exhibits useful properties, such as antiatherogenic properties and sufficient mechanical strength in stent devices. However, pure zinc has some limitations, including lower corrosion rates in vivo and relatively inferior mechanical characteristics [136,139].

Surface treatments of zinc-based biomaterials are needed to regulate their biodegradation rate [122]. To improve performance, different alloy elements are integrated into zinc, increasing the liquidity of the molten metal and improving the mechanical properties of the alloy [140]. In orthopedic surgery, zinc-based biomaterials are designed for long-term retention in order to achieve their intended purpose, as relatively low biodegradation rates appear. However, the over-release of Zn^2+^ during the decomposition process can lead to cytotoxic effects in the laboratory and delay bone integration within the body [122].

#### 2.1.3. Nanocrystalline Metallic Materials

The orthopedic field has witnessed significant efforts to develop enhanced biomaterials for a range of applications. These applications encompass temporal osteosynthesis implants utilized in the treatment of bone fractures or critical defects, as well as permanent implants such as total knee replacement prostheses [141,142]. While bulk properties, specifically mechanical resistance and load-bearing capacity, are essential for bone-substituting implants [143,144,145], surface-related properties play a critical role in achieving optimal performance. These properties include osseointegration, osteosynthesis performance, infection prevention, and corrosion resistance [141].

Titanium oxide coatings have been extensively studied due to their demonstrated biocompatibility and excellent osseointegration in dental implants, which can be attributed to the native oxide of Ti-based implants [141,146,147,148]. However, concerns have arisen regarding the extensive use of Ti-based and TiO_2_ materials, including their limited bioactivity, reduced corrosion resistance in media containing F^−^ or Cl^−^ over prolonged periods, and the resulting adverse effects of titanium accumulation on the human body [149,150,151]. Reports of allergic reactions and hypersensitivity associated with titanium necessitate research into alternative materials [152,153].

Other biocompatible transition metal oxides have exhibited promising biological properties, including osseointegration, enhanced cell adhesion and proliferation, reduced inflammatory response, antibacterial effects, and remarkable corrosion and wear resistance [154,155,156,157]. However, there has been limited research into the biological response of potential oxides such as Nb_2_O_5_ and Ta_2_O_5_ [158,159,160,161]. Another viable alternative is zirconium oxide (ZrO_2_), which offers suitable mechanical strength, corrosion resistance, and a favorable biological response for intraosseous applications [162,163].

### 2.2. Polymers

The word “polymer” is a Greek word derived from “poly” and “meros”, meaning “many” and “parts”, respectively [164,165]. A polymer comprises several reprised subunits. Natural and synthetic polymers play vital and ubiquitous roles due to their diverse functions in daily life [166]. Synthetic and natural polymers are the current and future biomaterials for orthopedic devices and bone tissue engineering. With the progress in technology, they can imitate the natural extracellular matrix (ECM) [2,167].

A polymer was employed as a biomaterial by coincidence when surgeons noted that World War II pilots who were injured by fragments of polymethylmethacrylate (PMMA) due to cockpit damage did not suffer any established deleterious reactions from the presence of Perspex shards inside their eyes [2,168,169,170]. Since then, PMMA has been employed in various medical applications. PMMA was the first synthetic polymer employed as a base for dentures in 1939 [171] and corneal replacement in the 1940s [172].

The new generation of degradable and bioresorbable medical implant biomaterials does not have any poisonous or mutagenic impacts. However, they do have some problems, such as limitations in strength and mechanical stiffness, unfavorable tissue responses, foreign body reactions, late tissue degradation reactions, and the potential for infection due to their crystallinity and hydrophobicity [173].

Polymers are classified into two main categories: synthetic and natural. Each section is further divided into two parts: biodegradable and non-biodegradable [174,175,176]. Figure 4 illustrates the categorization of polymers, including some examples.

#### 2.2.1. Natural Polymer

Natural polymers were employed as the first biodegradable biomaterials in medical applications due to their improved biological performance, excellent biodegradability, and high chemical versatility compared to traditional synthetic materials [2,177]. Hyaluronic acid, chitosan, collagen, gelatin, silk, cellulose, and alginate are among the most commonly used natural polymers. Specifically, chitosan, collagen, and chitin are the predominant natural polymers employed in medicine, particularly in the field of bone tissue engineering [2,167].

#### 2.2.2. Synthetic Polymer

##### Non-Biodegradable Synthetic Polymer

Most non-biodegradable synthetic polymers are biologically inert [178]. These substances were developed to minimize the host’s response to the biomaterial, aiming to reduce it to the lowest possible level. They serve as the foundation for numerous medical applications, including fracture fixators and orthopedic implants. Despite their customizable mechanical characteristics and high biological inertness, orthopedic implants made from non-biodegradable synthetic polymers and non-biodegradable orthopedic cement often experience high failure rates due to issues at the interface. These issues can arise from infections, poor integration with the surrounding tissue, or bone resorption caused by stress shielding [176].

i.Poly (Methyl Methacrylate) (PMMA)

Poly (methyl methacrylate) (PMMA) is extensively utilized as a nonmetallic implant material in orthopedics and bone grafting, specifically for fixing orthopedic prosthetics in the shoulders, knees, and hips. It is a non-biodegradable polyacrylate that possesses properties such as stiffness, thermos-plasticity, biological inertness, hydrophobicity, and biocompatibility. PMMA can be obtained through solution polymerization or the polymerization of methyl methacrylate using emulsion or mass methods. Its introduction into orthopedic surgery took place in the mid-1950s [179,180,181,182,183].

While bone cement based on PMMA is commonly employed and enables rapid primary stabilization of the bone, it lacks a biologically and mechanically stable interface with the bone. Additionally, it is prone to bacterial adhesion and the development of infections [184,185]. To modify treatment kinetics and impart mechanical properties, PMMA-based bone cement can be blended with bioactive glass or inorganic ceramics. In the 1970s, antibiotic-loaded PMMA cements were introduced to reduce the risk of prosthesis-related infections. However, self-curing PMMA cements have significant drawbacks, including their non-biodegradability, monomer toxicity, and the potential for necrosis of surrounding tissues due to high curing temperatures. Furthermore, PMMA cements exhibit limited interactions with the surrounding bone [185,186]. Figure 5 illustrates an X-ray image of a total knee arthroplasty performed on the right knee of a 73-year-old woman. The procedure involved the use of cement (PMMA). This X-ray was taken after a six-month follow-up period, and the arrows indicate the presence of the cement in the image.

ii.Polyetheretherketone (PEEK)

Polyetheretherketone (PEEK) is a thermoplastic polymer known for its high strength-to-weight ratio [187], excellent thermal stability [188], resistance to chemical and biological attacks [189,190], high melting temperatures [191], and glass transition [192]. At room temperature, it is chemically inert and insoluble in most solvents. There are two methods for producing PEEK: electrophilic reactions and nucleophilic displacement reactions, as illustrated in Figure 6 [193].

Despite its chemical resistance and biocompatibility, PEEK is biologically inert, resulting in limited interaction with the surrounding bone tissues and a potential risk of implant failure. Several studies have explored strategies to transform PEEK into a bioactive material and enhance its compatibility with bone implants [194,195,196,197,198,199,200,201]. One approach involves modifying the surface of PEEK, while another approach focuses on creating bioactive compounds based on PEEK [202].

When a biomaterial is implanted in vivo, water molecules are the first to reach the surface of the implant. Subsequently, proteins interact with the implant, a process influenced by the adsorbed water molecules. Following this, cells adhere to the adsorbed proteins, thereby impacting tissue growth [203,204]. However, PEEK’s low surface energy hinders cellular adherence since the implant initially interacts with water molecules. To promote cellular attachment and spreading, it is crucial to have a substrate with high surface energy, creating a hydrophilic surface. Surfaces with higher energy stimulate faster cell attachment and spreading compared to surfaces with lower energy. Thus, modifying the surface energy of a polymer alters its surface reactions, resulting in an optimal surface for the intended application [205,206].

Wang et al. employed plasma immersion ion implantation to modify the surface of PEEK. In addition, hydroxyl groups were grafted onto the PEEK surface to impart hydrophilic properties to the modified material. The cytocompatibility of both pure PEEK and the modified PEEK was also evaluated. The study revealed that the modified surface significantly enhances osteoblast adhesion, spreading, and proliferation, which can accelerate bone maturation around the implant [207].

iii.Polythene (PE)

Polyethylene (PE) is a thermoplastic polymer that undergoes melting at a specific temperature and solidifies upon cooling in a reversible process. This characteristic allows for repeated cycles of melting and solidification without significant degradation. Due to this property, PE is widely used, enabling injection molding and rapid shaping into desired sizes and shapes. PE exhibits semi-crystalline behavior due to its symmetric molecular structure, which can lead to crystallization and subsequently affect density and chemical stability. Most types of polyethylene are chemically resistant, with only specific formulations being soluble in high-temperature solvents such as toluene, xylene, trichlorobenzene, or trichloroethane [208]. PE has a density ranging from 0.88 to 0.96 g/cm^3^, and varying molecular weights and branching. The American Society for Testing and Materials recognizes five major types of PE: low-density polyethylene (LDPE), linear low-density polyethylene (LLDPE), medium-density polyethylene (MDPE), cross-linked polyethylene (XPE), and high-density polyethylene (HDPE) [209].

Ultra-high-molecular-weight polyethylene (UHMWPE) is a specific type of polyethylene (PE) that is linear and semicrystalline. It has a high molecular weight. In orthopedics, UHMWPE typically has a molecular weight ranging from 3.5 to 6 million g/mol and a crystallinity degree of 50–55% [210]. Worldwide, approximately three million bone joint replacement surgeries are performed annually, with the majority involving the use of UHMWPE implants [211]. UHMWPE has an elastic modulus closer to that of bone compared to other commonly used prosthetic materials such as Ti–6Al–4V and Co–Cr–Mo metal alloys (Table 5). Prostheses with a significantly different elastic modulus than natural bone can lead to stress shielding, resulting in a reduced mechanical load on the bone due to the bone–implant interaction, which may lead to bone loss around the implant [183].

##### Biodegradable Synthetic Polymer

Compared with natural polymers, biodegradable synthetic polymers possess superior mechanical properties and thermal stability. Extensive research has been conducted on the group of synthetic (α-hydroxy) polymeric biomaterials. Among them, the most frequently employed polymers are polyglycolic acid, polylactic-co-glycolic acid, and polylactic acid. These polymers are capable of biodegradation or absorption under in vivo conditions, making them ideal matrices for applications in regenerative medicine [2,212].

i.Poly (Glycolic Acid) (PGA)

Since the 1970s, polyglycolide acid (PGA) has been the pioneering biodegradable polymer used for orthopedic fixation, specifically in screws, nails, and plates. It is a robust and dense polymer with a molecular weight ranging from 2.0 × 10^4^ to 1.45 × 10^5^ g·mol^−1^, a glass transition temperature (Tg) of 35 to 40 °C, and a melting point (TM) of approximately 224 °C [3]. PGA exhibits a highly crystalline structure, but its degradation time is relatively short, lasting from 6 to 12 months. Due to its rapid degradation properties, PGA demonstrates an early decline in mechanical strength in vivo, occurring approximately 4 to 7 weeks after the implantation procedure. Moreover, the use of PGA in bone surgery for orthopedic fixation can potentially lead to adverse side effects such as sinus wound formation, swelling, and fluid accumulation, primarily due to the increased presence of glycolic acid [213].

ii.Poly (Lactic Acid) (PLA)

Polylactic acid (PLA) is derived from natural organic lactic acid and belongs to the category of biodegradable thermoplastic aliphatic polyesters [214]. PLA possesses several distinct qualities that make it an environmentally and economically attractive biopolymer, including its excellent rigidity, superb transparency, excellent processability, and glossy appearance. However, it does have some limitations, such as intrinsic brittleness, poor toughness, and a slow degradation rate [215]. Nevertheless, PLA has shown great potential as a biomaterial in various medical applications, including regenerative medicine, orthopedics, and tissue engineering. It has also gained significant importance as a printable biopolymer for 3D printing [216]. PLA is particularly well-suited for bone fixation devices, including absorbable plates and screws. By manipulating the polymer’s stereochemical structure and molecular weight, it is possible to adjust the ratio and degree of crystallization, thereby influencing its mechanical properties, degradation behavior, and processing temperatures. PLA exists in two stereoisomers: poly (L-lactide) (PLLA) and poly (D-lactide) (PDLA). Due to the racemic combination of monomers, PDLA is amorphous, leading to a disruption of crystallinity and consequently faster erosion compared to PLLA. In vitro and in vivo comparisons between PLLA and PDLA under physiological conditions have shown that highly crystalline PLLA may take two to five years to degrade, while amorphous poly (D,L-lactic acid) loses its integrity within two months and completely degrades within twelve months [217]. As a result, polymeric biodegradable plates and screws have been employed in various surgical procedures, including maxillofacial surgery, pediatric surgery, and orthognathic surgery (Table 6) [213].

iii.Poly (Lactide-Co-Glycolide)

This copolymer combines the desirable characteristics of PGA and PLLA, including adjusted degradability, biocompatibility, and hydrophilicity [2,218]. PLGA is hydrolytically degraded to the principal acidic constituents, lactic and glycolic acids, which are biologically removed to avert complications. However, these acidic degradation products, present at higher concentrations in PLGA interference screws, can alter the behavior of osteoblasts. These molecules impede cell proliferation and accelerate differentiation, thereby hindering the healing process around the degrading polymer implant in vivo. To mitigate the release of acidic by-products, PLGA has been combined with ceramics such as beta-tricalcium phosphate and hydroxyapatite [2,219,220].

iv.Poly (Caprolactone)

PCL is a stiff, aliphatic, semi-crystalline, and biodegradable non-toxic polyester. It demonstrates sufficient biocompatibility and can be categorized into three groups based on a single glass transition temperature (Tg) to determine mechanical compatibility. Notably, poly (ε-caprolactone) stands out with its low melting temperature (59–64 °C) and low glass transition temperature (60 °C) [2,221,222,223]. These properties make PCL suitable for various applications, such as orthopedics, tissue engineering scaffolds, drug delivery systems, and sutures [224]. However, PCL does have some limitations, including its slow degradation process, which can take up to 3 or 4 years, and its hydrophobic nature, which hampers cell adhesion and penetration [175]. Nevertheless, PCL offers advantages over polyhydroxy acids (PHA) due to its cost-effectiveness, wide availability, and greater stability. By employing copolymerization or blending it with other polymers, the properties of PCL can be modified [224].

## 3. Enhanced Bone–Implant Biocompatibility due to Osteogenic Factors

While minerals, ceramics, and polymers are available to meet diverse industry needs, the biomedical sector has witnessed the significant adoption of metals and metal alloys for medical implant applications. However, despite their unique mechanical properties, metals do not exhibit favorable biocompatibility when implanted in the body. Researchers have expressed concerns regarding the potential toxicity effects of various implant metals, including titanium, aluminum, stainless steel, and iron. These metals possess a high modulus, which can result in stress shielding, triggering inflammatory or foreign body reactions. Additionally, they have been associated with various infections, such as yellow bile syndrome caused by titanium ions and osteomalacia due to the leaching of aluminum ions, among others [225].

Modern biomaterials have revolutionized our understanding of the intricate interactions occurring within biological systems, operating at both the cellular and molecular levels. The ultimate objective of these investigations is to develop materials and products that are better suited for diverse applications in biomedicine. Significant progress has been made in recent years in creating such materials through the utilization of nano-coatings, grafted polymer brushes, nanotubes, hydrogels, and organic and inorganic nanoparticles, among others. Polymer materials often undergo modifications to enhance their biological properties. These modifications encompass various components, such as drugs and biomolecules absorbed on or loaded into the scaffold’s surface, incorporation of inorganic micro- and nanoparticles onto or within the scaffold, as well as the application of coatings onto the scaffold’s surface. These advancements have been introduced to address the aforementioned concerns [226,227].

Osteogenic factors can help enhance bone–implant biocompatibility when implant toxicity is a concern. Implant toxicity refers to the adverse effects caused by the materials used in the implant, which can impede the healing process and integration with the surrounding bone. Osteogenic factors can mitigate these toxic effects and promote better biocompatibility. Examples that illustrate this concept are as follows:

A study was conducted to evaluate the biocompatibility and osteogenic potential of various calcium-silicate-based cements (CSCs) when combined with an enamel matrix derivative (Emdogain) using human-bone-marrow-derived mesenchymal stem cells. The results demonstrated that the incorporation of CSCs as retrograde filling materials and the administration of additional Emdogain resulted in significant improvements in bone regeneration and prognosis for apical microsurgery procedures [228].

Concentrated growth factors (CGF) refer to a product obtained from autologous blood through the centrifugation of venous blood. In a study, researchers investigated the potential of CGF to stimulate the osteogenic differentiation of human bone marrow stem cells (hBMSC) in an in vitro setting. The findings unequivocally demonstrated that CGF alone possesses the ability to induce osteogenic differentiation of hBMSC [229].

Another study focused on the utilization of exosomes, which are nano-sized extracellular vesicles containing proteins, nucleic acids, and lipids, as therapeutic nanoparticles for treating diseases. The researchers developed a cell-free tissue engineering system by employing functional exosomes instead of seed cells. They achieved this by constructing gene-activated engineered exosomes that encapsulated the VEGF gene derived from ATDC5 cells. Effective integration of the engineered exosomes with 3D-printed porous bone scaffolds was accomplished using a specific exosomal anchor peptide known as CP05. The study demonstrated that these engineered exosomes play a dual role: they act as an osteogenic matrix, inducing the osteogenic differentiation of mesenchymal stem cells, and also function as gene vectors, allowing controlled release of the VEGF gene to remodel the vascular system [230].

A study was conducted to investigate the potential of 3D-printed β-tricalcium phosphate (β-TCP) scaffolds in promoting the osteogenesis of bone marrow stem cells (BMSCs) through N6-methyladenosine (m6A) modification. The results revealed that β-TCP exhibited excellent biocompatibility and demonstrated osteoinductive properties. Furthermore, the study observed an increase in methyltransferase-like 3 (METTL3), which resulted in an elevated m6A level of RUNX2. Consequently, this led to a more stable level of RUNX2 mRNA [231].

In a conducted study, researchers aimed to develop two self-assembling supramolecular hydrogels by utilizing an osteogenic growth peptide (OGP) and evaluate their impact on proliferation and osteogenesis in both in vitro and in vivo settings. The hydrogels, known as F-sequence and G-sequence hydrogels, exhibited remarkable biocompatibility and demonstrated the ability to enhance cell proliferation. Additionally, the hydrogels effectively stimulated the upregulation of crucial osteogenic factors, including RUNX2, BMP2, OCN, and OPN, thus promoting the process of osteogenic differentiation. The findings of this study provided significant insights into the underlying mechanism involved in hydrogel-mediated repair of bone defects [232].

## 4. Additive Manufacturing (AM)

Additive manufacturing, also known as 3D printing, is a manufacturing technique that creates 3D objects by adding material layer by layer. It is considered the third pillar of overall manufacturing technology, alongside subtractive manufacturing techniques such as milling or lathing, and formative manufacturing techniques such as casting or forging. Additive manufacturing was formerly known as rapid prototyping (RP) and is now commonly referred to as 3D printing. In 1984, Chuck Hull developed stereolithography, the first additive process for polymers, which was later commercialized. He coined the term “stereolithography” and patented the technology in 1986 [233,234]. There are various additive manufacturing techniques, including fused-deposition modeling, 3D inkjet printing, stereolithography, direct powder extrusion, and selective laser sintering. These techniques involve digitally controlled layer-by-layer deposition of materials to create different geometries of printlets [235].

Additive manufacturing has emerged as a prominent research topic in the past decade due to its low cost, ease of use, and the reliability of 3D printing equipment. It enables the straightforward and customized production of complex 3D structures and components through the layer-by-layer deposition of materials, eliminating the need for specialized tools or molds [234,236].

Additive manufacturing offers a unique opportunity for fabricating personalized dosage forms, which is crucial in addressing the diverse medical needs of patients. Despite having been in existence for four decades, AM has only recently gained wider usage in both surgical and non-surgical fields [48].

Furthermore, 3D printing is becoming a popular method of producing medical devices for orthopedic applications, tissue engineering, and the rehabilitation of patients suffering from disabling neurological diseases such as spinal cord injuries and amyotrophic lateral sclerosis. This is due to 3D printing enabling the creation of patient-specific designs, highly complex structural elements, and affordable on-demand manufacture [237]. Moreover, human organs can be manufactured according to the principles of AM using specialized 3D bioprinters [238]. Techniques of 3D printing have great potential for fabricating porous, complex-formed substances, and forms with highly complex internal structures. As a result, 3D printing technology allows the creation of hierarchical substances with mechanical qualities (strength and elastic Young’s modulus) and porous structures comparable to natural bone, while reducing the stress-shielding impact created by orthopedic implants [239,240,241].

Although 3D printing enables researchers to create parts that meet these requirements, the majority of clinical work in orthopedics focuses on metallic biomaterials, and most commercial representation is centered around metal-related approaches. However, polymers and polymeric composites receive significant attention in bone engineering applications due to the strong similarities between their thermomechanical properties and those of tissues, as well as their biodegradability and biocompatibility [242]. Furthermore, 3D printing technologies offer several advantages, including mass production capability, economic efficiency, and repeatability [243]. Moreover, when combined with computer-aided design (CAD) [244], 3D technology can be used to create completely patient-specific implants [245,246].

Despite the significant advances that have been made in 3D printing technology, there are still notable problems to overcome. These include software design, standardization and integration of a comprehensive bio-fabrication platform, repeatability, limitations of 3D printers’ capabilities, biomaterial characterization, regulatory hurdles, and quality by design. Addressing these challenges is crucial for 3D printing to be recognized as a traditional bio-manufacturing method in medicine and to gain access to the medical market. Among these challenges, the lack of heterogeneous biomaterials that would enable their reliable clinical utilization is the main obstacle [237].

### 4.1. Additive Manufacturing of Metallic Implants

Over the years, porous metal biomaterials have been fabricated using conventional manufacturing techniques, primarily based on powder metallurgy, such as metal injection molding and spacer processes. Although these fabrication techniques have made remarkable progress, certain limitations still exist, including the inability to precisely control pore shape and distribution, as well as dimensional inaccuracies [247]. Additionally, implants manufactured using conventional processing methods, known as standard-type implants, cannot match the structure, performance, and required physical and chemical characteristics for addressing specific bone flaws. This limitation restricts the therapeutic efficacy and longevity of implants [52].

On the other hand, additive manufacturing (AM) of metallic biomaterials demonstrates excellent medical potential, encompassing prostheses, implants, drug delivery systems, scaffolds, and stents. The process of AM involves creating and manufacturing 3D designs, which can be achieved using a CAD program. Furthermore, these 3D designs can be manufactured using various techniques such as fused deposition modeling (FDM), selective laser melting (SLM), and selective laser sintering (SLS) [248].

AM technology offers several advantages, including more precise fabrication and greater flexibility in designing both the internal and external macro- and micro-architectures of orthopedic implants [249]. The geometrical and topological porosity qualities of metallic biomaterials can be accurately tuned through controlled AM manufacturing techniques, improving their mechanical properties to mimic bone [250,251]. This leads to improved rates of bone tissue regeneration [252,253,254], altered biodegradation kinetics [255,256], and the formation of a vast, interconnected osteocyte lacuno-canalicular network [257,258].

However, certain characteristics such as wear resistance, hardness, anti-ferromagnetic properties, or antibacterial characteristics cannot be easily modified through geometrical design alone, as these properties require modifications to the underlying base material(s) prior to AM processing [247].

### 4.2. Additive Manufacturing of Polymeric Implants

Polymers play a crucial role in 3D printing manufacture due to their versatility, excellent processability, and compatibility with various AM processes [48,259]. They possess notable characteristics such as surface detailing, high precision, temperature resistance, accuracy, and improved strength [260]. In the realm of additive manufacturing, polymers contribute significantly, accounting for 51% of the polymer parts produced, 29% of metal and polymer combinations, and 19.8% of metal products [261]. Reactive monomers, thermoplastic filaments, powder, and resin are commonly utilized forms of polymers in AM techniques [259].

Despite the wide array of available AM techniques, advancements in polymer printing primarily focus on three key strategies: (1) powder bed fusion processes such as selective laser sintering (SLS), (2) deposition-on-demand processes, including extrusion-based technologies such as fused deposition modeling (FDM) and direct-ink-write printing [262], as well as inkjet or drop-wise deposition methods, and (3) photo-polymer-based printing techniques, such as stereolithography (SLA). These printing methods have successfully incorporated various polymers as raw materials [259,263].

The production of polymer composites through 3D printing entails both advantages and drawbacks, with each method having specific requirements regarding the polymer’s structure, state (liquid or solid), and physical characteristics (melting temperature and viscosity) [264]. When it comes to load-bearing applications, the range of materials that can be employed for polymer AM is comparatively limited compared to other applications. For instance, commonly used polymers in SLS processes for load-bearing applications include PEEK, UHMWPE, PMMA, PLA, PCL, polypropylene (PP), polyvinyl alcohol (PVA), and polyamide (PA) [48].

## 5. Future Direction

The field of medical science is advancing rapidly to meet the increasing demand for biomedical implants used in the treatment of injuries and traumas [265]. This progress is driven by the development of new materials and the integration of three-dimensional fabrication techniques, enabling the production of complex geometric implants [227]. However, to ensure the safety and efficacy of these implants, it is crucial to conduct toxicity analysis and biocompatibility studies on the alloying elements used in metallic implants that undergo controlled degradation.

The interfacial properties of implants in a biological environment can be improved through the utilization of advanced surface modification techniques and coatings. Understanding the chemistry of coating materials, conducting corrosion analysis, and optimizing fabrication techniques and parameters are essential for achieving efficient and accurate outcomes.

The development of novel composite materials holds great potential in enhancing the mechanical performance of implants, providing flexibility, and introducing new functionalities. Non-biodegradable implants present challenges such as stress shielding, toxicity, and the generation of harmful byproducts. To address these issues, biodegradable implants are being developed, which support tissue growth and self-sustainability. Future research into biodegradable materials aims to focus on strategies for controlling impurity levels, optimizing coatings and alloying elements, functionalizing implant materials, optimizing the biodegradation rate at the implant/tissue interface, and exploring new degradable materials.

In tissue engineering, the design of polymeric scaffolds is being optimized to elicit timely and desirable responses. The demand for tissue and organ replacements drives advancements and efforts in tissue engineering [266]. Polymeric scaffold materials offer the ability to control various physical and chemical properties for tissue engineering applications [267]. However, further research and development are needed to overcome limitations related to porosity, bioactivity, and mechanical properties. The study of a wide range of materials and fabrication techniques is crucial in addressing these limitations.

Overall, the continuous exploration of new materials, fabrication techniques, and surface modifications is essential for the advancement of biomedical implants and tissue engineering. Through interdisciplinary research and collaboration, the field can overcome the existing limitations and pave the way for safer, more efficient, and effective implant solutions [266].

## 6. Conclusions

Bone fractures present complex challenges for scientists and orthopedic surgeons. In orthopedic surgery, both metals and synthetic polymers have been utilized and compared. Metal implants, such as titanium and its alloys, and stainless steel and its alloys are commonly used for fracture repair due to their excellent mechanical properties, strength, and toughness. However, these materials exhibit limited biocompatibility, leading to foreign-body interactions such as poisoning, inflammation, swelling at the surgical site, and high corrosion rates, often requiring additional surgeries for removal.

On the other hand, polymeric implants have emerged as an alternative to metal implants, offering high biocompatibility, non-toxicity, and biodegradability. However, their mechanical properties are comparatively poor. Polylactic acid is a promising biopolymer due to its unique qualities, including its excellent rigidity, transparency, processability, and glossy appearance. It is an ecologically and economically attractive material. To enhance the mechanical strength and biocompatibility of biodegradable polymer implants, it is necessary to develop controlled absorption rates and explore methods for increasing their mechanical properties.

Additive manufacturing plays a significant role in the production of scaffolds and orthotic devices. It allows for customized designs for individual patients, cost-effective on-demand manufacturing, and the creation of complex porous structures with internal features similar to natural bone. This technique enables the fabrication of graded materials with varying porosities and mechanical properties. Additionally, special 3D biological printers can be utilized to produce composite human organs, following the principles of additive manufacturing.

The field of orthopedic implants and tissue engineering is exploring a range of materials and manufacturing techniques to address the challenges associated with bone fractures. The use of biodegradable polymers, additive manufacturing, and advanced 3D printing technologies offers promising solutions for improving biocompatibility, mechanical properties, and customization in the development of implants and tissue engineering constructs.

## Figures and Tables

**Figure 1 polymers-15-02601-f001:**
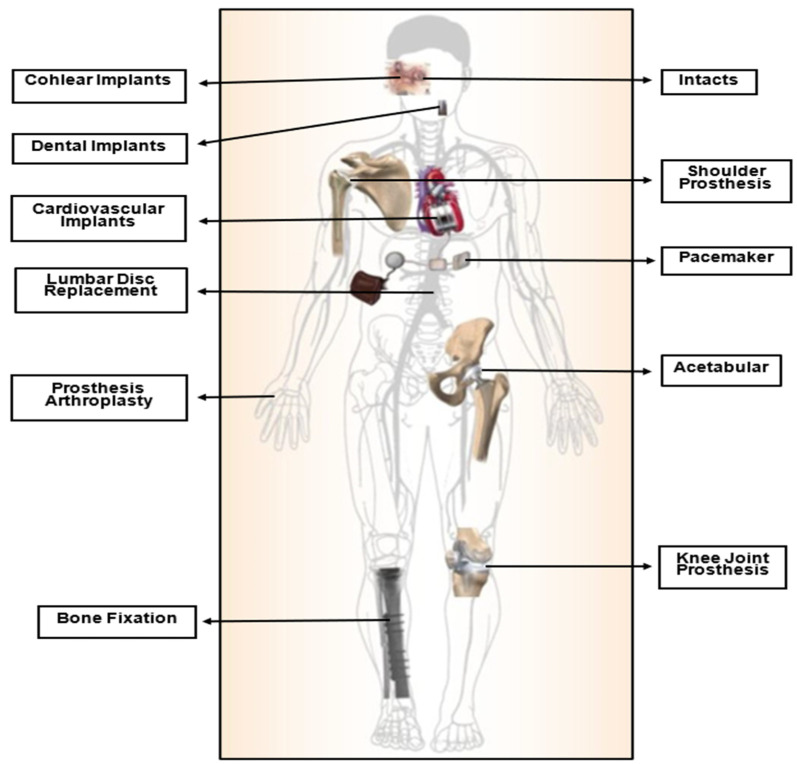
Bio-devices utilized in the human body [10].

**Figure 2 polymers-15-02601-f002:**
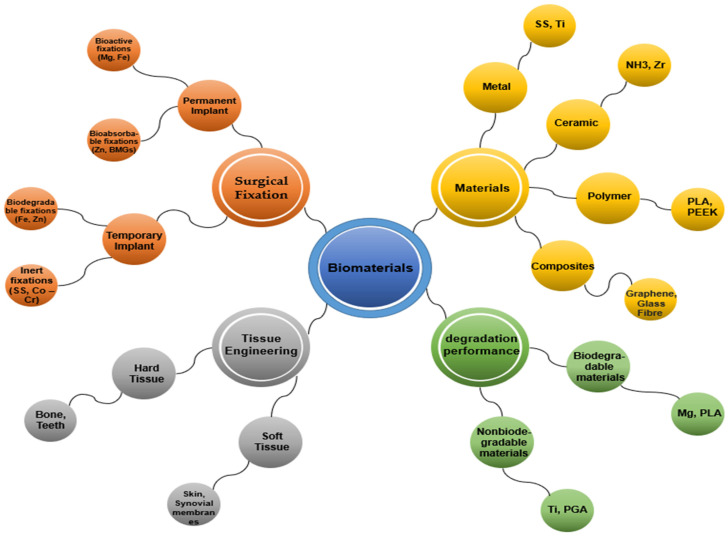
Classification of biomaterials.

**Figure 3 polymers-15-02601-f003:**
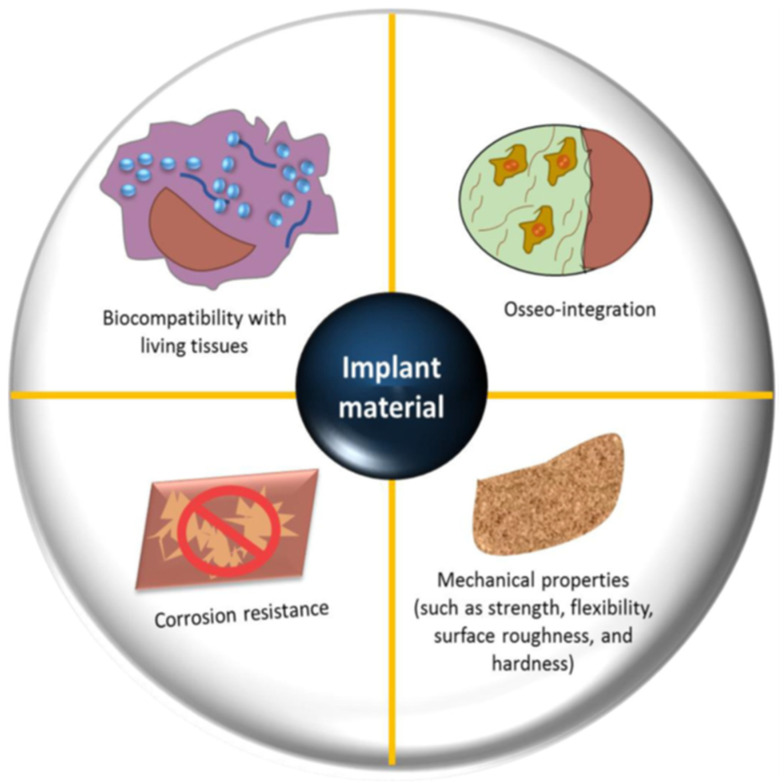
Factors that need to be considered in the selection and design of an orthopedic bio-implant [30].

**Figure 4 polymers-15-02601-f004:**
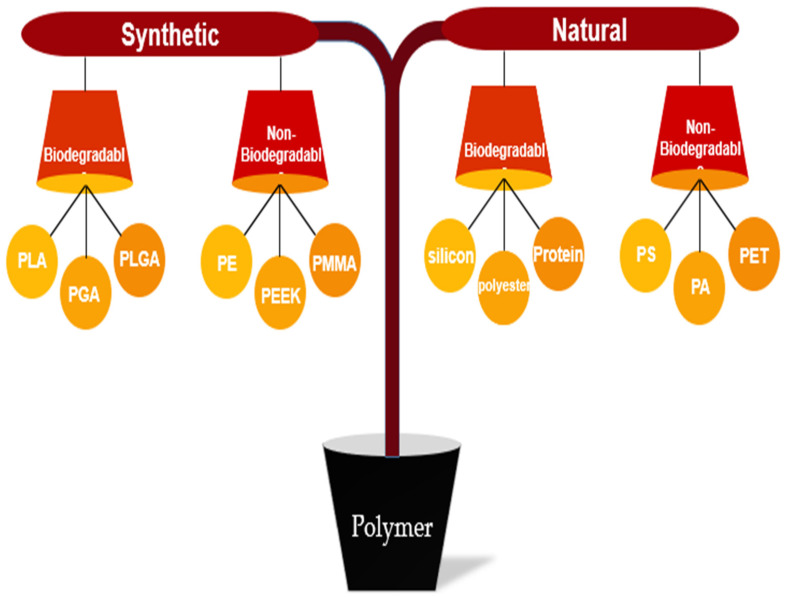
Classification of synthetic and natural polymer biomaterials.

**Figure 5 polymers-15-02601-f005:**
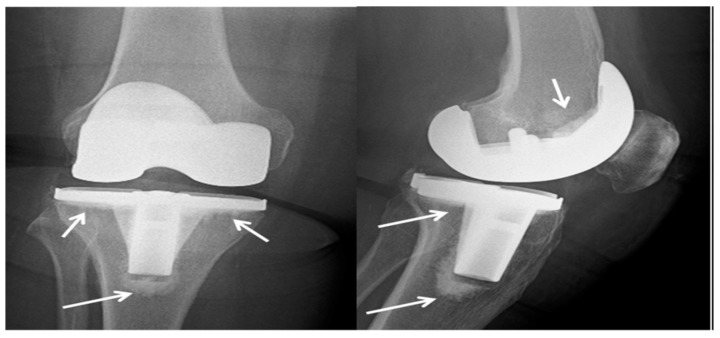
X-ray of total knee arthroplasty made from cement (PMMA) in the right knee of a 73-year-old woman after a six-month follow-up period, where the arrows point to the cement [184].

**Figure 6 polymers-15-02601-f006:**
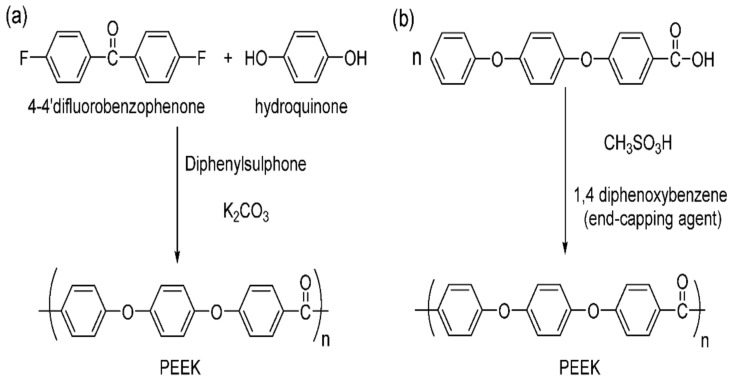
(**a**) Nucleophilic and (**b**) electrophilic polyetheretherketone structure [183].

**Table 1 polymers-15-02601-t001:** Locations of various human body fractures that demand different temporary fixation implants [14,38].

Fracture Site		Internal Fixators
Head	Fracture of the skull	Plates, wires, and pins
	Craniofacial fracture	Plates, wires, and pins
Trunk	Fracture of the clavicle	Plates and intramedullary nails
	Fracture of the scapular	Plates and screws
	Fracture of the pelvis	External fixators, screws, and plates
	Spinal fracture	Fixation implant contains plates, pedicle screws, and rods
Upper limb fracture	Fracture of the humerus	Plates and screws for open reduction and an intramedullary nail for closed reduction
	Fracture of the radius or ulnar	Plates and screws for open reduction and an intramedullary nail for closed reduction
	Fracture of the phalanges and metacarpal fracture	External fixators for close reduction and intramedullary nails, plates, and screws for open reduction
Lower limb fracture	Femoral fracture	Plates and screws for open reduction and an intramedullary nail for closed reduction
	Tibial and fibular fracture	Plates and screws for open reduction and an intramedullary nail for closed reduction
	Fracture of the metatarsus	Plates and screws for open reduction and an intramedullary nail for closed reduction
	Calcaneal fracture	Wires and screws for close reduction

**Table 2 polymers-15-02601-t002:** Mechanical characteristics, biocompatibility, and corrosion resistance of metallic and polymeric biomaterials that are extensively utilized in orthopedics.

Materials			Density (g/cm^3^)	Yield Strength (Mpa)	Tensile Strength (Mpa)	Elongation at Break (%)	Elastic Modulus (Gpa)	Biocompatibility	Corrosion-Resistance	Refs.
Metal	Non-biodegradable	316 L steel	7.9	290	579	40	193	Poor	Reasonable	[50,51,52,53]
		Ti-6Al-4V	4.43	850–900	960–970	14	110	Fair	Excellent	
		CoCr20Ni15Mo7	7.8	240–450	450–960	50	195–230	Poor	Excellent	
	Biodegradable	Pure Mg	1.74–2	65–100	90–190	2–10	41–45	Excellent	Poor	[50,53]
		Fe20Mn alloy	7.73	420	700	8	207			
		Zn-Al-Cu (Zn based alloy)	5.79	171	210	1	90			
Polymer	Non-biodegradable	UHMWPE	0.931–0.949	21.4–27.6	38.6–48.3	3.5–5.25	0.894–0.963	Good	Excellent	[54,55]
		PMMA	1.18	-	72	5	310	Good	Excellent	[56,57,58,59,60]
		PEEK	1.23–1.32	87–95.2	70.3–103	0.3–1.5	3.76–3.95	Good	Excellent	[61,62,63,64]
	Biodegradable	PLA	1.21–1.25	60	21–60	6	0.35–3.5	Excellent	poor	[2,65]
		PLGA	1.30–1.34	3.8–26.6	13.9–16.7	5.7	-	Excellent	Poor	[2,50]
		PLC	1.11–1.14	8.37–14.6	20.7–42	22.8–28.3	0.21–0.44	Excellent	Poor	

**Table 3 polymers-15-02601-t003:** Diverse types of bio-metal materials employed in orthopedic implants with their applications, advantages, and disadvantages [85].

Metal and Alloys	Particular Alloys	Major Applications	Advantages	Disadvantages
Stainless steel	316 L Stainless steel	Surgical implements, stents, fracture fixation	High wear resistance	The modulus is increased compared to bone allergy due to Co, Cr, and Ni
Titanium-based alloys	CP–Ti	Dental implants, fracture fixation, bone and joint replacement, pacemaker encapsulation	Low density, excellent biocompatibility, high corrosion resistance, low Young’s modulus	Weak tribological characteristics, the toxic impact of V and Al with long-term use
	Ti–Al–Nb			
	Ti–6Al–4V			
	Ti–13Nb–13Zr			
	Ti– Mo–Zr–Fe			
Co and chromium alloys	Co–Cr–Mo	Dental implants and restorations, heart valves, joint and bone replacement	Excellent wear resistance	The modulus is increased compared to bone allergy due to Co, Cr, and Ni
	Cr–Ni– Cr–Mo			
Others	Ni–Ti	Orthodontic wires, fracture fixation plates, stents	Low Young’s modulus	Allergy due to Ni
	Platinum Pt–Ir	Electrodes	Excellent corrosion resistance under maximum voltage potential and charge transfer conditions	-
	Hg–Ag–Sn amalgam	Dental restorations	Easily moldable in situ into a desired shape that is resistant to corrosion in the oral environment	Concerns related to Hg toxicity

**Table 4 polymers-15-02601-t004:** Physical properties of metallic alloys compared to human cortical bone [116].

Material	Young’s Modulus (GPa)	0.2% Offset Yield Point (MPa)	Compressive Strength at 20% Strain (MPa)
Cast Fe	203	157	498
Cast Mg	30–40	20–30	100–180
Cast Zn	100	95	200
Cast Fe–35Mn	-	240	440
Human cortical bone	1–35	1–20	103–140

**Table 5 polymers-15-02601-t005:** The differences in elastic modulus values between bone and materials commonly used in prosthetic manufacturing [183].

Material	Elastic Modulus (GPa)	Tensile Strength (MPa)
Trabecular bone	0.02–0.05	1–5
Cortical bone trabecular	3–30	50–151
UHMWPE	0.9–2.7	50–151
PMMA	1.88–3.3	68
PEEK	3.5–4.0	118
Co–Cr–Mo alloy	210–232	1173
Ti–6Al–4V alloy	116	1018

**Table 6 polymers-15-02601-t006:** The system of polymeric biodegradable implants, including plates and screws, utilized for bone fixation [213].

Product Name	Manufacturer	Polymer Composition	Degradation Time
Biofix^®^ SR-PGA	Bionx Implants, Tampere, Finland	SR-PGA	6 weeks
Biofix^®^ SR-PLLA	Bionx Implants, Tampere, Finland	SR-PLLA	5–7 years
Resomer^®^ LR708	Evonik Industries, Darmstadt, Germany	PLLA (70%) + PDLLA (30%)	2–3 years
MacroPore^®^	MacroPore Biosurgery Inc., San Diego, CA, USA	PLLA (70%) + PDLLA (30%)	2–3 years
Macrosorb^®^	MacroPore Biosurgery Inc., San Diego, CA, USA	PLLA (70%) + PDLLA (30%)	2–3 years
Biosorb FX^®^	Linvatec Biomaterials Ltd., Tampere, Finland	PLLA (70%) + PDLLA (30%)	2–3 years
Resorb X^®^	KLS Martin Group, Tuttlingen, Germany	PLLA (50%) + PDLLA (50%)	12–30 months
PolyMax^®^	Synthes, Oberdorf, Switzerland	PLLA (70%) + PDLLA (30%)	2 years
PolyMax^®^ RAPID	Synthes, Oberdorf, Switzerland	PLLA (85%) + PGA (15%)	12 months
Rapidsorb^®^	DePuy Synthes, West Chester, PA, USA	PLLA (85%) + PGA (15%)	12 months
Lactosorb^®^	Lorenz, Jacksonville, FL, USA	PLLA (82%) + PGA (18%)	12 months
Delta^®^	Stryker Leibinger Corp., Kalamazoo, MI, USA	PLLA (85%), PGA (10%), PDLA (5%)	8–13 months
Inion CPS^®^	Inion Inc., Tampere, Finland	PLLA, PGA, TMC–proportion varies	2–4 years
Inion CPS^®^ baby	Inion Inc., Tampere, Finland	PLLA, PGA, TMC–proportion varies	2–3 years
Osteotrans-MX^®^	TEIJIN Medical Corp., Osaka, Japan	PLLA (60–70 wt%), u-HA (30–40 wt%)	4.5–5.5 years

SR, self-reinforced; PGA, polyglycolic acid; PLLA, poly-l-lactic acid; PDLLA, poly-d-l-lactic acid; TMC, trimethylene carbonate; u-HA, unsintered hydroxyapatite; USA, United States of America.

## Data Availability

Not applicable.

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
