# Peer review of "Biomaterials as Implants in the Orthopedic Field for Regenerative Medicine: Metal versus Synthetic Polymers"

_polymers, 2023, doi:10.3390/polym15122601_

Round 1
Reviewer 1 Report
This review compares metallic and synthetic polymer implant biomaterials as implants in the orthopaedic field for regenerative medicine. This is an exceptionally well-done work. And I have no hesitation to recommend publication in this Journal.
I have just a couple of suggestions:
1. Some minor errors need to be fixed. Such as:
Line 161 "the majority of the ..."
2. It is preferable to add descriptions of metal oxides and nanocrystalline metal.
Moderate editing of English language
Author Response
|
Num. |
Comments |
Action taken |
|
1. |
Line 161 "the majority of the ..." |
The required correction in this comment has been done. Modified into" The majority of the...'' and replaced the line from 161 to 66. |
|
2. |
It is preferable to add descriptions of metal oxides and nanocrystalline metal. |
The required correction in this comment has been done. Added '' section 2.1.3. Nanocrystalline Metallic Materials''
|
|
3. |
Moderate editing of English language |
The required corrections in this comment have been made using the Grammarly app and ChatGPT. |
Reviewer 2 Report
This is a review on "Biomaterials as implants in the orthopaedic field for regenerative medicine: Metal versus Synthetic Polymer". The review is accurate and actual. However, throughout the all document the term Biomaterials is used both as Biomaterials for all tissues and biomaterials for bone tissue.
The authors should start to speak about Biomaterials in general and then focus on bone tissue related biomaterials, Please reorganize that in order the work to be published,
Also Page 1 Line 35 it is "Bone is an living tissue" and not "Bone is in living tissue"
Author Response
|
Num. |
Comments |
Action taken |
|
1. |
The review is accurate and actual. However, throughout the all document the term Biomaterials is used both as Biomaterials for all tissues and biomaterials for bone tissue. |
The required correction in this comment has been done. Add '' Section 2.1.3. Nanocrystalline Metallic Materials'' , ''Section 3. Enhance Bone-Implant Biocompatibility by Osteogenic Factors'' and Section 5. Future Direction'' |
|
2. |
The authors should start to speak about Biomaterials in general and then focus on bone tissue related biomaterials, Please reorganize that in order the work to be published, |
The required correction in this comment has been done. And I reorganized my work again as follows: 1- Combining Section 1. Introduction, section 2. Biomaterial mechanical, corrosion and biocompatibility characteristics and Section 3. Classification of biomaterials''. 2- Divide it into two sections'' Section 1. Introduction and 2. Biomaterials In Orthopedic'' 3- Delete ''Table 1. Historic milestone development of biomaterials for orthopedic implants.'' 4- Replace ''Figure 2 with Figure 3 and vice versa.'' 5- '' Section 2.1.3. Nanocrystalline Metallic Materials'', ''Section 3. Enhance Bone-Implant Biocompatibility by Osteogenic Factors'' and Section 5. Future Direction''. |
|
3. |
Also Page 1 Line 35 it is "Bone is an living tissue" and not "Bone is in living tissue". |
The required correction in this comment has been done. Modified into" Bone is a living tissue...'' and replaced the line from 35 to 126. |
Reviewer 3 Report
While the review contains interesting information, it is presented in a dry, textbook-like manner that is unlikely to capture the attention of readers.
To make it more engaging, it should include specific examples of successful implant usage with detailed explanations. Additionally, some important information, such as the significance of Figure 2 in the selection and design of orthopedic biomaterials, is not adequately explained. Similarly, the text barely touches on the differences between the two parts presented in Figure 5 and why it's important to understand them.
To address these issues, the review should include a separate section on implant toxicity and provide more information on osteogenic factors that can enhance implant biocompatibility.
Furthermore, a section discussing future trends and approaches to improve implant properties would be beneficial. This would enable readers to gain insight into upcoming trends in a concise and comprehensive manner.
Finally, I suggest citing the paper: https://doi.org/10.3390/ma14061417
Minor editing of English language required
Author Response
|
Num. |
Comments |
Action taken |
|
1. |
While the review contains interesting information, it is presented in a dry, textbook-like manner that is unlikely to capture the attention of readers. |
I reorganized my work again. |
|
2. |
To make it more engaging, it should include specific examples of successful implant usage with detailed explanations. Additionally, some important information, such as the significance of Figure 2 in the selection and design of orthopedic biomaterials, is not adequately explained. Similarly, the text barely touches on the differences between the two parts presented in Figure 5 and why it's important to understand them. |
The required correction in this comment has been done. Replace ''Figure 2 with Figure 3 and vice versa.'' and Add '' Every biomaterial must fulfill multiple essential criteria, such as possessing suitable mechanical properties (e.g., precise weight and elastic modulus), demonstrating good biostability (resistance to hydrolysis, oxidation, and corrosion), ensuring biocompatibility, particularly in the case of bone prostheses (promoting osse-ointeGration), exhibiting high bio-inertness. (non-toxic and non-irritant characteris-tics), demonstrating high wear resistance, and enabling easy application in practical settings [30], [38]–[40], see Figure 2.'' While the Figure 5.'' According to the source, the author did not provide any further information regarding Figure 5.'' And Add '' Figure 5 illustrates an X-ray image of a total knee arthroplasty performed on the right knee of a 73-year-old woman. The procedure involved the use of cement (PMMA). This X-ray was taken after a six-month follow-up period, and the arrows indicate the presence of the cement in the image.'' |
|
3. |
To address these issues, the review should include a separate section on implant toxicity and provide more information on osteogenic factors that can enhance implant biocompatibility. |
The required correction in this comment has been done. Add '' Section 3. Enhance Bone-Implant Biocompatibility by Osteogenic Factors''. |
|
4. |
Furthermore, a section discussing future trends and approaches to improve implant properties would be beneficial. This would enable readers to gain insight into upcoming trends in a concise and comprehensive manner. |
The required correction in this comment has been done. .Add '' Section 5.Future Direction''. |
|
5. |
Finally, I suggest citing the paper: https://doi.org/10.3390/ma14061417 |
The required correction in this comment has been done. The references have been placed, as [229] |
|
6. |
Require minor English editing. |
The required corrections in this comment have been made using the Grammarly app and ChatGPT. |
Round 2
Reviewer 3 Report
The authors have answered to all my issues, the paper can be accepted in its present form.
Minor editing of English language required.